# Exploring Private Sector Engagement for Faecal Sludge Emptying and Transport Business in Khulna, Bangladesh

**DOI:** 10.3390/ijerph18052755

**Published:** 2021-03-09

**Authors:** Shirish Singh, Ankita Gupta, Muhammed Alamgir, Damir Brdjanovic

**Affiliations:** 1Water Supply, Sanitation and Environmental Engineering, IHE Delft Institute for Water Education, Westvest 7, 2611 AX Delft, The Netherlands; d.brdjanovic@un-ihe.org; 2A408, Pradhan Urban Live, Bharat Nagar, Bhopal 462039, India; ankitagupta137@gmail.com; 3University Grants Commission of Bangladesh, UGC Bhaban, Dhaka 1207, Bangladesh; alamgir_member@ugc.gov.bd; 4Department of Biotechnology, Delft University of Technology, Julianalaan 67, 2628 BC Delft, The Netherlands

**Keywords:** faecal sludge, emptying and transport, enabling environment, financial analysis, private sector

## Abstract

In Khulna, Bangladesh, mechanical faecal sludge (FS) emptying and transport (E&T) service is provided by community development committees (CDCs) and the Khulna City Corporation (KCC). Without considering capital expenditure and depreciation, financial analysis for one year revealed that a CDC-1 m^3^ vacutug made a profit of Bangladeshi taka (BDT) 145,780 (USD $1746) whereas a KCC-2 m^3^ vacutug was in the loss of BDT 218,179 (USD $2613). There is a need to engage the private sector for sustainable service provision. Some of the key elements of enabling the environment for private sector engagement are policy/strategy, institutional and regulatory framework, implementation capacity, and financial viability. Existing policy/strategy/frameworks acknowledged the need and suggested plans for private sector engagement, and decentralised authority to city corporations. With increasing private-public partnership projects and collaboration in the sanitation sector, capacity of the KCC and the private sector are increasing. Financial viability of the FS E&T business is primarily dependent on the number of trips and the emptying fee. For the E&T business to be financially viable, a 2 m^3^ vacutug should make six trips/day (internal rate of return (IRR)—13%, discount rate—6.5%) with an emptying fee of BDT 750 (USD $9)/m^3^. Despite the lack of operative guidelines for faecal sludge management (FSM), enabling the environment for private sector engagement in FS E&T business in Khulna seems favourable.

## 1. Introduction

Globally, more than 2.7 billion people are reliant on-site sanitation systems (OSSs), predominantly, septic tanks and pit latrines, and the number is expected to rise to 4.9 billion by 2030 [1]. A mixture of human waste, water, and any other materials are disposed of and stored in pits, tanks, or vaults of OSSs is faecal sludge (FS). FS should be properly contained and safely managed to minimise environmental pollution and reduce public health risks. Faecal sludge management (FSM) is a systematic approach of dealing with FS, which typically follows five components of sanitation service/value chain (i) containment (ii) emptying (iii) transport (iv) treatment (v) reuse and disposal. The current need of FSM worldwide, particularly in lower and middle-income countries, is large and inequitably distributed amongst the poorest households in both rural and urban areas [2]. Effective FSM service delivery requires proper containment of OSS, emptying of OSSs, and subsequent transport to treatment and safe disposal [3], which will ensure that FS is safely collected and disposed, and not simply relocated from OSS to the immediate environment. Emptying and transport (E&T) of FS is an important component in FSM, which is provided by different stakeholders, e.g., private sector, informal sector, Non-Governmental Organisations (NGOs), local government, etc. In major cities across Africa and Asia, the most common emptying technologies constitute mechanised and semi-mechanised technologies [4]. Some of the mechanised technologies include conventional vacuum tanker, cesspool truck, and vacutug, whereas semi-mechanised technology options include manual pit emptying technology (MAPET) and Gulper [5]. In most developing countries, there is a lack of governing policies or regulatory framework on FSM, and FS E&T services are often provided without adequate technology, regulations, and safety precautions [6].

Bangladesh has an estimated population of 166 million in 2018 and about 36.6% living in urban areas [7]. Urban Bangladesh has officially been declared open defecation-free and OSS is the prevailing practice in urban areas. As per the World Health Organisation (WHO) and the United Nations Children’s Fund (UNICEF) Joint Monitoring Programme (2019) data, 50.7% of the urban population has basic sanitation services, 31.8% have limited service, and 17.5% have unimproved sanitation services. In absence of the expansion of sewerage networks (due to high costs), Bangladesh is facing a second-generation or post-ODF challenge [8], and there is dire need for FSM. In 2017, the Institutional and Regulatory Framework for FSM (IRF-FSM) was published and disseminated by the Policy Support Branch of the Local Government Division (LGD) under the Ministry of Local Government, Rural Development and Cooperatives. The framework outlines how FSM services can be implemented, as well as the roles and responsibilities of different institutions and stakeholders. The guideline “Occupational Safety and Health Guidelines for faecal sludge management” aims to minimise risks involved in septic tank/pit emptying, transportation, and disposal of FS to the greatest extent possible and provides guidance for protection of workers and the environment [9].

Khulna is the third largest industrial city, located in the southwest region of Bangladesh, comprising of 31 wards. The city is 45.65 square km in area, inhabiting approximately 1.5 million people (66,257 holdings) [10]. Khulna City Corporation (KCC) is a formation under the local government administration of Bangladesh to regulate the city area of Khulna; it has been collaborating with stakeholders and initiated several activities for improving FSM services, including E&T of septic tanks/pits and the FS Treatment Plant (FSTP). FSTP was built to demonstrate FSM with a future plan to cover the entire city; however, it is not even receiving the minimum load to run properly, which is due to the fact that OSSs are not properly constructed and there is inadequate demand and collection of FS E&T services. Currently, E&T services are being provided by the conservancy department of KCC and three Community Development Committees (CDCs), a committee formed under the Urban Partnership for Poverty Reduction project. KCC is collaborating with local stakeholders to improve OSSs and planning to implement scheduled FS desludging to increase FS load in the FSTP. Current FS E&T services are inadequate at meeting the demands, and existing E&T businesses are questionable, due to financial sustainability without any external grant. It has been realised that, engaging the private sector in E&T service provision could leverage private funding, contribute in meeting FS E&T demand, better services and, consequently, FSM.

Despite a large number of public-private partnership (PPP) projects under the central government of Bangladesh, in areas of energy, transport, port, water supply, etc., very few PPP projects are under the local government, and most of them are confined to solid waste [11]. A study by Water & Sanitation for the Urban Poor (WSUP) revealed that PPPs offer a strong platform for developing approaches to FSM in Bangladesh [12]. It could allow local governments to provide FSM services in an efficient and cost-effective way, ensuring services are sustainable, reducing costs and risks for both parties, and utilising the private sector’s strength in innovation, business expertise, and quality services. For FS E&T, several PPP models in Bangladesh are being implemented. One example is the SWEEP model—the very first private mechanised FS E&T service, implemented in Dhaka and replicated in Chittagong, where a vacuum tanker is leased out to a private party to provide FS E&T services. It has been profitable and successful [12]. In this model, the local government/authority purchases a vacuum tanker and leases it to a private party, undertakes infrequent maintenance, marketing, regulation, deals with FS disposal, and increases the vacuum fleet, whereas the private party pays a security deposit and a monthly lease fee to the local government/authority, provides emptying services, and undertakes regular operations and maintenance of the tanker. Since smaller-scale PPPs are being implemented, there is a need to develop a separate regulatory framework for such PPPs.

This study is limited to the FS E&T component of FSM. The aim of this study is to (i) understand practices and challenges of FSM, particularly related with E&T; (ii) carry out financial analysis of existing E&T businesses; (iii) explore private sector engagement in E&T businesses in Khulna. All subjects involved in the study gave their informed consent for inclusion prior to participating in the study.

## 2. Materials and Methods 

This study was conducted through an extensive desk study focused on FSM in Khulna, the private sector engagement for FS in Bangladesh, and a two-month field study in Khulna. Field data collection methods (household survey, key informant interviews, etc.) were conducted in the local language, which was further translated and transcribed in English.

A comprehensive desk study, by reviewing various literature and publications, was conducted to understand existing FSM practices, particularly on E&T, its challenges, key stakeholders involved in FSM, and enabling the environment for private sector engagement. Interviews were conducted with relevant stakeholders along with field observations to have a deeper understanding of existing practices and challenges. Convenience sampling was adopted for interviews, as there were limited FSM stakeholders. A list of semi-structured interviews and key interview topics are listed in Table 1.

A random sample household survey was conducted to ascertain the types of FS containment and its sizes, emptying frequency, emptying service provider, emptying tariff, and willingness to pay. The survey was conducted in ward no. 17 (Sonadanga area), which is largely representative of the entire city, as this is a mix of planned development, unplanned development, and slum areas, as well as community of manual sweepers known as “Harijans” living in the ward. The required sample size for household survey was calculated using the following formula (https://www.surveymonkey.com/mp/sample-size-calculator/ accessed on: 17 November 2020):(1)$$Sample\;size=\frac{\{{(z}^{2}\times p\left(1-p\right)\}÷e^{2}}{\left.1+[\left\{z^{2}\times p\left(1-p\right)\right\}÷e^{2}N\;\right]}$$
where, z = standard deviation for a desired confidence level; p = population proportion; *e* = margin of error; *N* = population size.

Since the main aim of the survey was to triangulate/validate secondary data, which were used to assess the potential market of the E&T business, a confidence level of 90% and margin of error 10% was adopted. For a population size of 30,352 in ward no. 17, and z = 1.65 for 90% confidence level, the required sample size is calculated as 68 (70 adopted).

Financial data of current E&T businesses (KCC and CDC) were collected from February 2018 to January 2019 (one year). A template to collect financial data, which comprised of capital expenditure, operational expenditure (office expenses, personnel, fuel, maintenance, and other costs), and revenue generated by the business was developed. To start with, the template was shared with concerned financial staff of KCC and CDC ward no. 17, who were requested to fill out the template. A one-to-one meeting was then organised with them to fill out any missing data, and to clarify/confirm the data. Financial data were analysed in Excel to determine internal rate of return (IRR), which is a standard method to calculate an investment’s rate of return. For any business to be financially viable, the IRR should be higher than the discount rate. Interviews were also conducted with key staff of these businesses to have a clear picture of service provision and associated problems/issues.

## 3. Results and Discussion

### 3.1. Containment

Although open defaecation is a rare phenomenon, a substantial percentage, 18% of the population, rely on shared toilets and 7% on unimproved toilets. A baseline study conducted in 2014 revealed that toilets were connected to two major types of containment systems—septic tanks (61.7%) and pits (29.3%) [13], whereas this study revealed that 76% of households had septic tanks with/without soakwell, and 24% had pits as containments. It was found that the average volume of a septic tank was 16.64 m^3^ and pit was 1.96 m^3^. The volume of septic tank was similar to the average volume (16.2 m^3^) found by the study by Chowdry and Kone, 2012; however, the volume of the pit was smaller than the volume (3.2 m^3^) found in the same study [4].

### 3.2. Emptying and Transport (E&T)

It is the responsibility of households to empty their containment; it is generally emptied when it starts overflowing or when there is a foul smell. Approximately half of the households surveyed emptied their containment at least once in less than three years. This is similar to average emptying frequency in Phnom Penh (once in three years) and Siem Reap, Cambodia (once in two years) [14]. The emptying frequency varied, with some households emptying their containment in less than one year, while some households did not empty in 15 years. Higher emptying frequency was attributed to smaller pit volumes and high water table, whereas lower emptying frequency was because the effluent from the containment discharged in open drains, which resulted in public health risks. Scheduled desludging could be a potential solution to minimise public health risks (due to groundwater pollution and disposal into open drains) and to increase and steady demand of E&T services. At the same time, it might be easier for households to pay for emptying services on a regular basis (monthly payment) rather than a one-time payment when emptying, particularly for low-income households. In the survey, it was discovered that willingness to pay for emptying services regularly, on a monthly basis, was preferred rather than a one-time payment when emptying. Approximately 77% said that they were ready to pay in the range of Bangladeshi taka (BDT) 50–250 (USD $0.6–3.0) per month.

Both manual and mechanised emptying services are available in Khulna. Manual emptying is provided by sweepers (informally) whereas mechanised emptying services are provided by KCC and CDC (formally). Manual emptying is performed generally in pits, using simple tools, such as buckets, spades, and ropes. Working in groups of 2–3, one emptier often climbs into a pit to empty it, while other(s) pass filled buckets to empty the contents nearby without using any personal protective equipment (PPE) [15]. For manual emptying, households directly call sweepers, whereas for mechanised emptying, the household has to either call CDC or book vacutug services at the KCC office by purchasing a booking voucher, and pay an emptying fee upfront via a “*challan*” form. Approximately 70% of households surveyed responded that they prefer mechanised cleaning over manual cleaning, primarily due to the foul smell during manual emptying and objection from neighbours. Regarding emptying service providers, households preferred services of KCC to CDC because services provided by KCC are cheaper and they are easier to contact (many households are unaware of services provided by CDC). However, the emptying service of KCC is not timely, and in many cases, households have to use CDC services since emptying is done when the pit/tank is full and starts overflowing.

KCC has two units of 2 m^3^ capacity vacutugs, and one larger vacutug of a capacity of seven m^3^, whereas three CDCs, from ward no. 03, 17, and 22 have a vacutug of 1 m^3^ capacity. A KCC vacutug consists of a steel tank mounted on a vehicle, which is connected to the vacuum pump operated by the gasoline engine of the vehicle (Figure 1). FS is pumped out from the tank/pit and could be discharged under gravity or by pressure. KCC vacutugs of 2 m^3^ capacity are imported, which costs around BDT 3,500,000 (USD $41,916), whereas seven m^3^ capacity is donated by an international non-governmental organisation (INGO). CDC vacutugs were donated by a United Nations (UN) agency in 2012, where every CDC was allotted 10 wards as their area of operation; however, they also operate outside their areas of operation, depending on availability of vacutug. Small sized vacutugs are predominantly used as they can access narrow lanes less than 2 m width, while larger vacutugs are used as mobile transfer stations. FS is transported to the FSTP located at a distance of around 10 km from the centre. The length of a trip depends on the location of the house; however, the longest trip of the vacutug, to and from (city-household-FSTP-city) is about 25 km.

### 3.3. Treatment

FSTP of capacity 180 m^3^/day is located at Rajbandh, south west of the city, adjacent to the landfill site of the city. The plant was established in 2017 and comprises of six drying beds of 54 m^2^ and six constructed wetlands of 900 m^2^ [16]. It is managed by an INGO. It was observed that the FSTP is not even receiving a minimum load of 30 m^3^/day because of improper septic tank design, illegal connection of outflow of the septic tank to open drains and leakages from the existing septic tank, as well as inadequate FS collection. Similar findings were obtained from discussions that existing septic tanks are not designed as per the Bangladesh National Building Code (BNBC), and are mostly constructed by masons without having proper knowledge of septic tank functions and design aspects.

### 3.4. Reuse and Disposal

Currently, there is no reuse of dried sludge and treated effluent. Treated effluent is discharged in adjacent drain of the FSTP site, whereas dried sludge is disposed of in a nearby landfill site.

### 3.5. Financial Analysis of Existing E&T Business

#### 3.5.1. 1 m^3^ Vacutug—CDC

Financial data of a 1 m^3^ vacutug of CDC ward no. 17 from February 2018 to January 2019 (one year) was collected and analysed, as shown in Table 2. It was recorded that during this period, a total of 208 applications were received to empty containments. The emptying fee was BDT 1000 (USD $12) per trip (per m^3^), and it required an average of 3.5 trips to empty a containment, which resulted in a revenue of BDT 728,000 (USD $8719) for the year. The emptying fee was higher compared to the emptying fee in several cities in Asia, but not as high as several cities in Africa [4,14,17]. Concerning operational expenses, it was categorised in four headings—(i) personnel; (ii) fuel; (iii) maintenance; and (iv) other expenses (communication, advertisement, and safety gears). Total expenses for the year was BDT 582,220 (USD $6973). Cash flow showed a profit of BDT 145,780 (USD $1746); however, it should be noted that capital expenditure and depreciation were not considered in the analysis.

Distribution of expenses into the four headings is presented in Figure 2. Personnel expenses are the highest (75%), followed by maintenance expenses (13%). It was surprising that fuel expenses (10%) was less than the maintenance expenses, which is due to the fact that the vacutug was old, which required higher maintenance costs. High maintenance requirements can consequently interrupt trips, which might affect the quality of E&T services.

#### 3.5.2. 2 m^3^ Vacutug—KCC

Similar to the 1 m^3^ vacutug, financial data for the 2 m^3^ vacutugs from KCC were collected and analysed. From records, it was revealed that a total of 156 applications were received from February 2018 to January 2019 (one year) for emptying containments. There is an application fee of BDT 10 (USD $0.12) and BDT 1500 (USD $18) emptying fee, which resulted in revenue of BDT 235,560 (USD $2821) for the year. Concerning operational expenses, it was categorised into four headings—(i) personnel; (ii) fuel; (iii) maintenance; and (iv) other expenses (communication, advertisement, and safety gears). Total expenses for the year was BDT 467,145 (USD $5595) and cash flow showed a loss of BDT 231,585 (USD $2774) even without considering capital expenditure and depreciation. This is primarily due to less emptying demand in the city. A study in Dakar, Senegal, has shown that profitability of emptying businesses can be improved by increasing tariffs, increasing the average number of trips per day, introducing fuel-efficient emptying tankers, etc. [18].

Distribution of expenses into the four headings is presented in Figure 3. Personnel expenses is the highest (69%), followed by fuel expenses (22%), which is similar to findings to FS E&T businesses in Thailand and Vietnam [19].

### 3.6. Enabling Environment for Private Sector Engagement

Sanitation and Water for All (SWA), which is a multi-stakeholder partnership, has recognised the need and importance for engaging the private sector in Water, Sanitation and Hygiene (WASH) and has developed a framework for enabling the environment, comprising three interlinked elements—(i) guiding principles; (ii) building blocks; and (iii) collaborative behaviour. All partners adhere to these elements of the framework, which contribute to multi-stakeholder dialogue, representing the values that partners share, their common understanding of the sector, what the sector needs to succeed, and how to meet these needs through collaborative action. At a country level, (i) sector policy strategy; (ii) institutional arrangements; (iii) sector financing; (iv) planning, monitoring, and review; and (v) capacity development are key elements of the building blocks that are critical for a well-functioning WASH sector [20]. In line with the SWA framework, the government of Bangladesh conducted a joint sector WASH bottleneck analysis (2018–2019) to assess the sector against SWA collaborative behaviours and building blocks in eight divisions; they took action regarding formulation, review, and updating of policies, strategies, and plans aligning the sustainable development goals. Two critical actions in FSM are—(i) establishment of an FSM support cell to address safely managed sanitation systems and (ii) a national action plan for IRF-FSM. The country has identified “encouraging private sector, local entrepreneurs, Micro Finance Institutions and external support agencies and NGOs to participate in WASH business in line with revised pro-poor strategy 2020”as part of the priorities and commitments [21].

Enabling the environment for private sector engagement is broad, and could include support for the private sector at a local level (e.g., access to financial support), as well as national policy/regulations and an international agenda [22]. At the local level, safely managed urban sanitation services/FSM require eight elements of enabling the environment—(i) policy, strategy, and direction; (ii) institutional arrangements; (iii) programme methodology; (iv) implementation capacity; (v) availability of products and tools; (vi) financing; (vii) cost-effective implementation; and (viii) monitoring and evaluation [23]. For this study, limited to FS E&T, enabling the environment for private sector engagement has been defined as conditions necessary for domestic business and entrepreneurs to operate and provide FS E&T services, and has considered following four elements—(i) policy/strategy; (ii) institutional and regulatory framework; (iii) implementation capacity; and (iv) financial viability.

#### 3.6.1. Policy/Strategy

The National Strategy for Water Supply and Sanitation (2014) lists promoting enhanced private sector participation as one of the guiding principles, along with the following strategies: (i) establish FSM, and (ii) facilitate private sector participation, which is highly relevant to engage private sector in FSM. The strategic directions needed to establish FSM system in relation with elements of enabling the environment are: allocating land for FS treatment and disposal; building FSM and regulation capacities of local government institutions; making arrangements, including bylaws for regular emptying of septic tanks and pit latrines, and providing technical and business support to the private sector in sludge management, recycling, and sale of compost or other products [24]. The strategy acknowledges benefits of private sector participation in the water supply and sanitation sector, which include mobilization of private resources to meet growing investment needs, reduction of cost of services through competition, and increased efficiency and innovation. The strategy has developed an implementation plan that includes preparation of guidelines for private sector participation in the sector; development of a package consisting of technical knowhow, business support, and financial assistance for private businesses in the sector, and others.

Local Government (City Corporation) Act 2009 mentions the responsibilities and functions of a City Corporation and clearly mentions, “City Corporation shall make adequate arrangements for collection and removal of refuse from all public streets, public latrines, urinals, drains, and all buildings and land within the jurisdiction of the city corporation”. Because “faecal sludge” was not widely used at that time, it is to be understood with the term “refuse” as referred to in the Act. For proper management of FS, if the city corporation deems it necessary, it could formulate necessary “rules”, “regulations”, and “by-laws”.

It is clear if the existing policy/strategy is quite favourable for private sector engagement in FSM, including FS E&T services.

#### 3.6.2. Institutional and Regulatory Framework

The primary objective of the Institutional and Regulatory Framework for faecal sludge management (IRF-FSM)—City Corporations is to facilitate implementation of FSM services in city corporation areas [25]. The framework defines specific roles and responsibilities of various institutions and stakeholders, particularly that of city corporations, for effective planning and implementation of FSM (including financial/business model for service delivery). Institutional roles and responsibilities are based primarily on provisions of the Local Government (City Corporation) Act 2009, which guides and regulates roles and responsibilities of all city corporations. City Corporations/LGD may formulate necessary rules, regulations, or by-laws (within framework of City Corporation Act 2009), if needed, for carrying out the specific roles and responsibilities outlined in the framework, and to initiate inclusive FSM planning and implementation modality among government agencies, I/NGOs, community groups, and the private sector.

Specifically, for FS E&T, the framework mandates City Corporation to carry out and/or oversee E&T, making sure that these operations are carried out in a hygienic manner without adversely affecting the health and safety of emptiers, the public, and the environment, as well as ensuring that collected FS is transported to the designated site(s) for treatment and disposal, and that collected FS is never disposed of in an open space, water bodies, storm drains, or sewers. It further specifies that City Corporation may engage private sector/non-government sector (e.g., outsourcing) for E&T, as a service procurement, and introduce and promote mechanised pit emptying (desludging) services. City Corporation may fix fees/charges for E&T of FS and may consider entire sanitation service/value chain (i.e., from collection to treatment) while fixing such fees/charges.

KCC is the key institution in planning, executing, and monitoring FSM. It is a regulating body of the city, which can formulate necessary rules, regulations, and by-laws for proper execution and monitoring of services in the city. It is also responsible for monitoring of discharge of FS from residential, commercial, and institutional areas at designated place. KCC monitors containment systems, provides mechanised E&T of FS, as well as oversees management of FSTP. CDCs provide mechanised FS E&T services, whereas manual sweepers are active for manual emptying of FS and cleaning the containment. NGOs (local and international) are engaged in all components of sanitation services/value chain, e.g., by raising awareness to eliminate open defaecation, improve existing sanitation facilities (individual as well as communal), technical and financial support to upgrade manual emptier professions (for better livelihood and dignity), and establishing FSTP (also supporting in operation and maintenance). These institutions are also striving to enhance capacity and develop business models for sustainability of FSM. Academic institutions/universities are helping to plan, design, and implement FSM by conducting research and translating research into practice. Representatives, i.e., ward councillors, are responsible for overall development in FSM and have mandates to enforce laws/regulations, as well as monitoring. The private sector is largely absent in the entire FSM.

Despite engagement of several stakeholders and good coordination, as well as clear definition of roles and responsibilities of institutions and stakeholders in the IRF-FSM, it was observed that there is a lack of operational guidelines on FSM.

#### 3.6.3. Implementation Capacity

KCC has been implementing several projects with private sector, regarding solid waste, road sector, and engaging private sector in sanitation. KCC issued a request for proposals and invited private sector to submit a technical and financial proposal for (i) emptying, collecting, and transporting of FS; and (ii) operation and maintenance of FSTP. At KCC, PPP are done on an ad-hoc basis, and there is a lack of a clear policy framework to work in the PPP model. It also lacks operative guidelines on FSM and the institutional monitoring system is inadequate. However, many efforts are being made to enhance KKC’s capacity on PPP, and have been quite successful to large extent.

There are many private sectors engaged in Khulna in several areas/sectors, but those related with sanitation are confined to selling and construction of toilets/containment. The study found that the private sector gained knowledge and awareness on FS E&T and was exposed to technologies for FS E&T. However, it also discovered some challenges of the private sector: (i) inadequate knowledge on financial transactions and viability of emptying business; (ii) lack of start-up capital; and (iii) uncertainty of the number of trips of emptying, which is directly related to profit. FS emptying (e.g., scheduled desludging) could be one of the key interventions in addressing the above challenges, which would assure a certain number of emptying trips, resulting in increased cash flow, i.e., profit. This would ultimately mean a financially viable business plan, supported by borrowing start-up capital from a local market/financial institution.

#### 3.6.4. Financial Viability of E&T Business

In order to engage the private sector for E&T services, E&T businesses should be profitable, and adequate emptying demand should be created. The study carried out a financial analysis for a 2 m^3^ vacutug to ascertain—(i) minimum number of trips required for the business to be financially viable and (ii) impact of the emptying fee on financial viability. In discussion with KCC and relevant stakeholders, it was assumed that scheduled desludging (once in three years) would be mandated, which will increase FS E&T demand. There is no collection or a plan to charge any discharging/tipping fees at FSTP, and licensing fees; hence, they are not considered in the financial analysis. Most of the data were collected in interviews and survey during the field study and some assumptions were taken from government records. The following data and assumptions were made to carry out the financial analysis [16]:Capital expenditure of vacutug—BDT 3,500,000 (USD $41,916);Operational expenditure:
○Personnel cost (one driver, two operators and one office staff)—BDT 60,000 (USD $719) per month;○Fuel cost—BDT 390 (USD $5.0) per trip;○Maintenance cost—0.5% of the capital expenditure;○Other costs (communication, advertisement, safety gear, office expens-es)—BDT 10,000 (USD $120) per month.
Maintenance cost increases by 15% per annum;Interest rate on capital expenditure is 6.5% per annum (discount rate);Depreciation—10% per annum;Emptying fee—BDT 1500 (USD $18) per trip (per 2 m^3^);Working days—300 days per annum;Revenue increases by 5.5% per annum [26].

Table 3 shows calculations of the financial analysis and reveals that, making six trips per day, the business could be financially viable (IRR—13%, discount rate—6.5%) with an emptying fee of BDT 1,500 (USD $18) per trip (per 2 m^3^). A substantial revenue generated by a high number of emptying services, low loan repayment due to low interest rate, and low maintenance expenses (as the vacutug is new) are some of the major reasons for positive net present value (NPV), and IRR higher than the discount rate. A similar analysis was done for emptying services of an NGO in Dhaka, which discovered that both NPV and IRR were negative, primarily due to a high loan repayment [27]. It has been established that higher emptying frequency and higher emptying fee results in increased profits of the emptying business [4]. The study found that, at the same emptying fee, IRR is negative (−1%) when it makes five trips per day (Figure 4). The analysis also showed that a change in +/− 10% in the emptying fee could have a significant impact on IRR, as seen in Figure 4. An increase of 10% in the current emptying fee could ensure that the business could be financially viable (IRR—9%, discount rate—6.5%), even with five trips per day. Considering an average desludging frequency of 3 years, a 1.96 m^3^ average pit volume, and willingness to pay in the range of BDT 50—250 (USD $0.6–3.0) per month, an emptying fee of BDT 1,650 (USD $20) per trip (per 2 m^3^) is even less than the minimum emptying fee that households are willing to pay.

## 4. Conclusions

It is evident that policy/strategy and institutional and regulatory frameworks have clearly acknowledged the benefits of private sector participation, which includes mobilization of resources to meet growing investment needs, reduction of costs of services through competition, and increased efficiency and innovation. Decentralization of authority empowers city corporations to formulate necessary “rules”, “regulations”, and “by-laws” to deliver FSM services. The study depicts that, a vacutug of 2 m^3^ capacity, with an emptying fee of BDT 750 (USD $9) per m^3^, requires at least six trips per day for businesses to be financially viable. It is also evident that financial viability of E&T businesses primarily depend on emptying fees and a number of trips per day. There should be adequate demand of emptying services, which could be achieved by introducing effective scheduled desludging and complying with the BNBC on septic tanks, monitored by KCC. It could be concluded that enabling the environment for private sector engagement in FS E&T businesses in Khulna is favourable; however, there is a need to develop FSM operative guidelines. The private sector could provide better services, but at the same time, could impose higher fees, if swayed by profit, which needs to be regulated by KCC.

## Figures and Tables

**Figure 1 ijerph-18-02755-f001:**
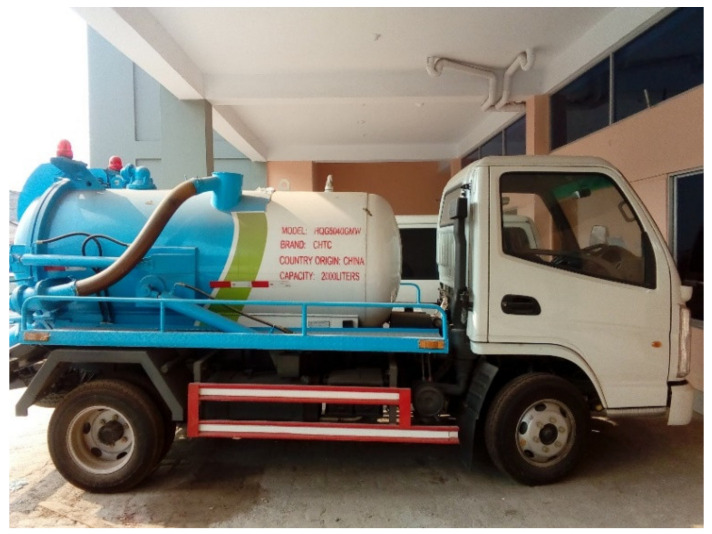
Khulna City Corporation (KCC) vacutug.

**Figure 2 ijerph-18-02755-f002:**
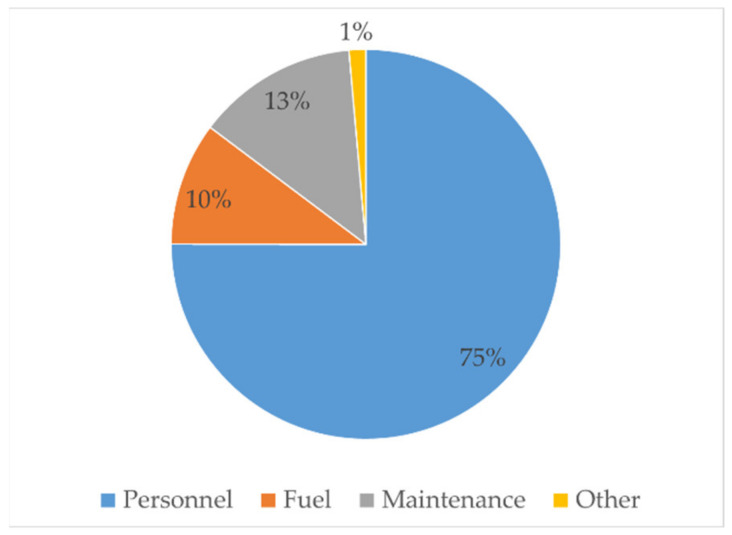
Distribution of expenses—1 m^3^ vacutug.

**Figure 3 ijerph-18-02755-f003:**
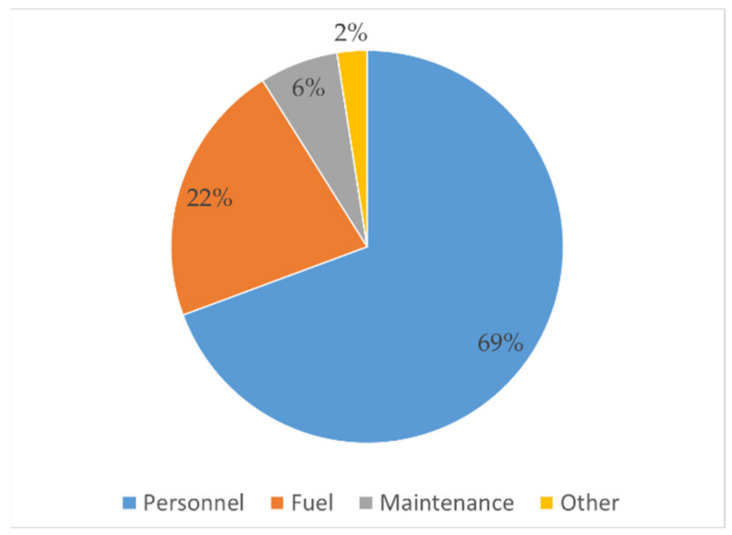
Distribution of expenses—2 m^3^ vacutug.

**Figure 4 ijerph-18-02755-f004:**
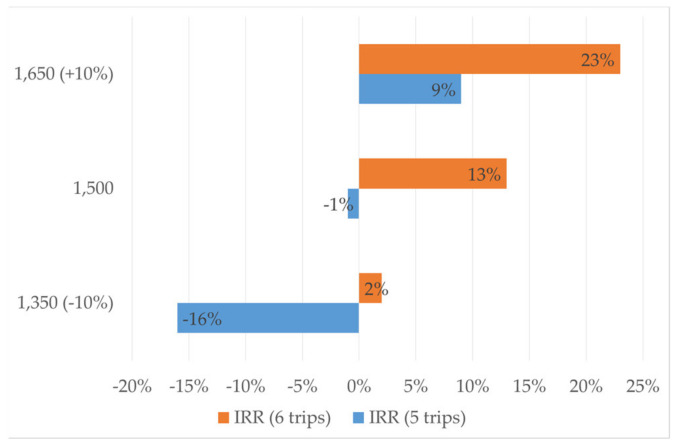
Impact of emptying fee on internal rate of return (IRR).

**Table 1 ijerph-18-02755-t001:** List of interviews and key interview topics.

Organisation	Position	Key Interview Topics
Khulna City Corporation (KCC)	Conservation officer Conservation inspector Revenue officer	Practices and challenges in FSM; Services and process for providing for FS E&T services; Revenue, capital, and operational expenditure of vacutug; Operation and maintenance of vacutug; Coordination with FSM stakeholders, particularly Community Development Committee (CDC).
International Non-Governmental Organisation (INGO)	City coordinator Governance advisor	Existing practices and challenges in FSM; Existing situation, operation, and challenges in FSTP; Awareness amongst stakeholders on mechanised FS E&T; Willingness to pay for desludging services.
CDC—Ward no. 3	Head of the cluster	Process of providing FS E&T service; Applications for FS E&T, emptying fee and revenue; Capital and operational expenditure of vacutug; Operational challenges; Access to funding for acquiring new vacutug; Mechanised emptying vs. manual emptying.
CDC—Ward no. 17	Head of the cluster	
CDC—Ward no. 22	Cluster leader	
Manual sweeper	Community leader	Challenges faced in manual emptying; Willingness to upgrade from manual to mechanised emptying; Reasons for engaging in this profession and social acceptance.
Manual sweeper	Representing middle aged group	
Manual sweeper	Representing youth group	

**Table 2 ijerph-18-02755-t002:** Current financial analysis of 1 m^3^ vacutug— community development committees (CDC) (February 2018 to January 2019).

**Revenue**														
Number of applications for emptying			208											
Average trips required per emptying (application)			3.5											
Emptying fee per trip			1000											
Revenue			728,000											
No.	Description	February 2018	March 2018	April 2018	May 2018	June 2018	July 2018	August 2018	September 2018	October 2018	November 2018	December 2018	January 2019	Cumulative
1	Revenue	60,667	60,667	60,667	60,667	60,667	60,667	60,667	60,667	60,667	60,667	60,667	60,667	728,000
**Expenses**														
Driver per trip			300											
Helper per trip			150											
Total personnel expenses (1 driver and 2 helpers)			436,800											
No.	Description	February 2018	March 2018	April 2018	May 2018	June 2018	July 2018	August 2018	September 2018	October 2018	November 2018	December 2018	January 2019	Cumulative
1	Personal expenses	36,400	36,400	36,400	36,400	36,400	36,400	36,400	36,400	36,400	36,400	36,400	36,400	436,800
2	Fuel expenses	7100	7750	8550	6150	3200	5150	3700	4800	1100	4950	4700	2400	59,550
3	Maintenance expenses	5150	2700	12,200	9,000	4,000	5,850	600	3,000	500	24,550		10,220	77,770
4	Other expenses—communication	300	300	300	300	300	300	300	300	300	300	300	300	3600
5	Other expenses—protective gears	1500	1500	1500										4500
	**Total cost**	50,450	48,650	58,950	51,850	43,900	47,700	41,000	44,500	38,300	66,200	41,400	49,320	582,220
**Cash Flows**														
No.	Description	February 2018	March 2018	April 2018	May 2018	June 2018	July 2018	August 2018	September 2018	October 2018	November 2018	December 2018	January 2019	Cumulative
1	Revenue	60,667	60,667	60,667	60,667	60,667	60,667	60,667	60,667	60,667	60,667	60,667	60,667	728,000
2	Expenses	50,450	48,650	58,950	51,850	43,900	47,700	41,000	44,500	38,300	66,200	41,400	49,320	582,220
3	Profit/loss	10,217	12,017	1717	8817	16,767	12,967	19,667	16,167	22,367	−5533	19,267	11,347	145,780

**Table 3 ijerph-18-02755-t003:** Financial analysis of 2 m^3^ vacutug.

**Capital Expenditure**												
1	Capital expenditure—2 m^3^ vacutug (loan)	BDT	350,0000									
2	Repayment period	years	10									
3	Interest rate	%	6.5%									
4	Equal Monthly Instalment (EMI)	BDT	39,742									
**Operational Expenditure**												
**No.**	**Description**		**Year 1**	**Year 2**	**Year 3**	**Year 4**	**Year 5**	**Year 6**	**Year 7**	**Year 8**	**Year 9**	**Year 10**
1	Personnel expenses		720,000	759,600	801,378	845,454	891,954	941,011	992,767	1,047,369	1,104,974	1,165,748
2	Fuel expenses		702,000	740,610	781,344	824,317	869,655	917,486	967,948	1,021,185	1,077,350	1,136,604
3	Maintenance expenses		17,500	20,125	23,144	26,615	30,608	35,199	40,479	46,550	53,533	61,563
4	Other expenses		120,000	126,600	133,563	140,909	148,659	156,835	165,461	174,561	184,162	194,291
5	Total		1,559,500	1,646,935	1,739,428	1,837,296	1,940,875	2,050,531	2,166,654	2,289,666	2,420,020	2,558,206
**Cash Flow**												
**No.**	**Description**		**Year 1**	**Year 2**	**Year 3**	**Year 4**	**Year 5**	**Year 6**	**Year 7**	**Year 8**	**Year 9**	**Year 10**
1	Revenue		2,700,000	2,848,500	3,005,168	3,170,452	3,344,827	3,528,792	3,722,876	3,927,634	4,143,654	4,371,555
2	Depreciation @10% of vacutug capital expenditure	0.9	350,000	315,000	283,500	255,150	229,635	206,672	186,004	167,404	150,664	135,597
3	Operational expenditure		1,559,500	1,646,935	1,739,428	1,837,296	1,940,875	2,050,531	2,166,654	2,289,666	2,420,020	2,558,206
4	Loan repayment		476,904	476,904	476,904	476,904	476,904	476,904	476,904	476,904	476,904	476,904
5	Net surplus	−3,500,000	313,596	409,661	505,335	601,102	697,412	794,685	893,313	993,660	1,096,067	1,200,847
6	Discount rate	6.5%										
7	Net Present Value (NPV)	1,531,715										
8	Internal Rate of Return (IRR)	13%										

## Data Availability

The data presented in this study are openly available in https://ihedelftrepository.contentdm.oclc.org/digital/ (accessed on 9 March 2021).

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
