# Peer review of "Exploring Private Sector Engagement for Faecal Sludge Emptying and Transport Business in Khulna, Bangladesh"

_ijerph, 2021, doi:10.3390/ijerph18052755_

Round 1

Reviewer 1 Report

I find the financial analysis rather basic, and the tipping fees not considered seem so vital to the whole system that the rest of the analysis is pointless without them. I think the authors need to rework the analysis with these included, and consider the model of the tip owners too.

There is not enough basic information for the general reader. I had to look up what a vactug was. Please include better description and a photo (there are a couple with creative commons licences if you do not have your own).

More description to get a proper feel for the private sector experience would be good – I am not clear for example how businesses are set up in the first place, where the loans are available (they seem very expensive -why?) and so on. The paper just scratches the surface of the enabling environment and more depth is needed.

The methods are also not clear. We have surveys and interviews and these are social research methods. We at least need the information about what was asked and how the answers were drawn out. It smacks of a social research programme being enacted by scientists or engineers who perhaps have not thought enough about the methodology, but maybe they have thought and just not said in the paper.

The term “cum” can’t be used. I detect you mean by it “cubic metres” so say that in full or use symbols such as 1 m3. “Cum” is a profanity in English and has to be removed.

The early part of the paper and the abstract need editing by someone with better English because the use of the definite article “the” is wildly incorrect. It gets better later on.

I hope the paper can be sufficiently revised because there is basic data in there somewhere that would benefit the research community.

Reviewer 2 Report

Multi-partner cooperation can help to improve the efficiency of public services, which is also a common practice in the international community. This paper studied the necessity and possibility of private sector engagement for faecal sludge emptying and transport business in Khulna, Bangladesh. The topic of the paper is interesting. However, there are some shortcomings that must be improved. Therefore, my current opinion is “Major revision”.

Comment 1

This paper discussed the practice of faecal sludge management (FSM) and analyzed some favorable environmental factors for private sector engagement. These factors mentioned in the paper are naturally essential for private sector engagement. However, the content of the paper is still incomplete. For example, what are the disadvantages or problems of private sector engagement in FSM? How to overcome these problems? The authors need to answer these questions.

Comment 2

The theoretical and innovative characteristics of this study are not prominent. It is recommended that the authors apply a conceptual framework or analysis tool in the revised paper.

Comment 3

The detailed discussions of some important issues are missing in this paper, such as the implementation mechanisms and operational guidelines of the system.

In addition, the structure of the paper is also unreasonable. It is suggested to optimize the paper structure and separate the part of "Discussions" from the part of "Results and Discussions" by following the basic  format of IJERPH.

Reviewer 3 Report

This study examines the status and context of faecal sludge management (FSM) services in Khulna, with a focus on the financial viability of sludge emptying and transport services. There has been significant effort from the sanitation sector to implement FSM services in Khulna in recent years, so research that provides insights on the success of these services is very welcome.

The study provides some interesting findings of revenue and expenditures, and preferences from sanitation users, that are worth publishing.

I think several major overall revisions to the manuscript are needed to improve the presentation of the information, fill in gaps in the methodology/discussion, or remove superfluous information:

  1. The manuscript would benefit from a more robust literature review on FSM as a business. There is a lot literature on this, including in Bangladesh (WSUP and SNV in particular have published findings on FSM businesses in Bangladesh). The authors do present some literature, but it is scattered across the results section instead of consolidated in a literature review section. Having some relevant literature on FSM entrepreneurs and businesses upfront, with discussion of the findings of this study in relation to the literature at the end, would strengthen the paper substantially.
  2. The methodology section requires much more detail on how each of the methods were designed and implemented. Currently, it’s very thin with only cursory mentions of interviews and observations (but not saying with whom or of what) and a bit on surveys.
  3. I don’t find the discussion on engaging the private sector to be particularly compelling. The authors state that engagement of the private sector is needed, but do not provide evidence from their study about why. Sections 3.6.1 and 3.6.2 are not terribly helpful because they simply summarise what is said in a couple policies and the IRF-FSM framework, but do not provide any new information, insights, or analysis. I think these two sections can be reduced in size. It is worth mentioning these policies, but I think this is more like background information to the study rather than a finding.

My suggestion would be to drop the language about supporting the private sector and focus on the financial viability of the public service providers and how they can be improved (or include a robust literature review of private FSM services providers and make an argument about how the findings in the study show that private sector engagement is needed with references from the literature. However, this would require much more work than simply dropping the private sector language).

My specific  comments are below.

Lines 17: I suggest that the authors write out “capital expenditure” instead of CAPEX.

Lines 19-22: I don’t think this study provides substantive evidence of why private sector engagement is needed (other than the presence of a policy that indicates support for it). There is not any evidence provided that a private service provider would fare any better than the public service providers, or provide services to the poorest communities (which the public service providers do). Further, the study shows that at least one of the public service providers is financially viable which indicates that a public service provider can work in the absence of private ones.

Line 35: The citations should be in numerical order. This first citation is [4] and subsequent citations seem to jump around randomly.

Lines 42-43: The sentence portion “Effective FSM service delivery requires proper containment of OSS, emptying of OSSs and subsequent transport to treatment and safe disposal…” seems to repeat what was just said in lines 37-40.

Line 49: I think the Vacutug is a fully-mechanised technology, not a semi-mechanised one. Semi-mechanised technologies like the Gulper require the user to power the device by hand. There are different versions of the Vacutug, but all Vacutugs use pumping/vacuuming technology which do not require manual power.

Lines 67-68: I think this sentence needs re-phrasing: KCC is not a city itself – it is a municipal local government body.

Lines 71-72: I think it would be helpful for the readers to provide a brief description of what KCC and Community Development Committees actually are. Also, I believe the E&T service is actually provided by the CDC Federation which is as organisation that oversees all CDCs in the city.

Line 77: Somewhere in this section, please provide details on what ethics approval was granted for this research.

Lines 82-83: Was this desk study about FSM focused on Khulna, Bangladesh, or globally?

Lines 84-87: There needs to be much more detail about the detail of the interviews and observations. Authors typically provide 1 or 2 paragraphs worth of detail for each method. For example, how many people were interviewed, which organisations they represented, were interviews structured or semi-structured, etc.

Lines 89-92: I think it is over-reaching to claim that this one ward is representative of the entire city. I am fairly certain that you would get significantly different results in other areas of the city.

Lines 92-94: What secondary data are you referring to here? Government data on sanitation status?

Line 96: You should spell out “Capital expenditure” and “operational expenditure” here.

Lines 100-101: Again, more detail is needed on how these interviews were designed and implemented.

Lines 105-106: Are the authors saying that, according to another study, 61.7% of users have a septic tank and 29% have a pit tank, but their survey found the proportions were 76% and 24%? It would be helpful to be clearer about where these reported figures are coming from.

 Line 108: I am used to seeing m3 for cubic meter instead of “cum”, but maybe this is just a stylistic choice. I think the authors can stick to whichever they prefer.

Lines 113-114: Do you mean they reported that they emptied their tank at least once every three years?

Lines 117-119: I feel like this is backward – smaller volumes and higher water tables would require higher emptying frequency (because they fill more quickly) and containment discharging to open drains would result from lower emptying frequency (because the containment is not being emptied frequently so it overflows).

Lines 123-126: What proportion of respondents stated they would prefer paying on a monthly basis versus a one-time payment? Also, what does a one-time payment mean? Does it mean they pay each time they get their containment emptied, or they make one payment and get emptying services indefinitely thereafter?

Lines 144-146: My understanding is that CDCs are at the ward level (or sub-ward level). As mentioned before, I think the actual operators are a higher level than the CDC (either CDC Federation or CDC Cluster).

Line 166: I think it would be helpful to clarify in this sub-heading that this is the CDC Vacutug.

Line 188: As above.

Line 269: Write out “service value chain” here instead of SVC (SVC has not been defined previously).

Line 275: What are “People’s representatives”? Are they elected officials (e.g. ward councillors)?

Lines 283-284: Why was the analysis only done for the 2m3 Vacutug and not the 1m3?

Lines 285-286: Can the author’s provide some background on how these assumptions were made? They appear to come from a Master’s thesis – perhaps this methodology can be summarised.

Reviewer 4 Report

A restructuring of the manuscript is needed. Clear results from the desk review, household survey, KII, Focus Groups, etc. should be separately described and detailed. There is limited connection between the methods and the results presented.

The household survey results should be clearly presented and the selection criteria more robust, otherwise no data can be represented. Sample size calculation should also be presented. Some, IRB approval should also have been gathered.

Under methods, the authors describe Ward No.17 as representative - please describe this assessment and determination.

The economic analysis reads like a business plan (justification) rather than an academic paper.

Round 2

Reviewer 1 Report

The authors have addressed most of my concerns except for the first point. I find the response here unacceptable. The authors state in their response that tipping was not collected in Khulna, yet in lines 304-386 they specifically say that it is not included and that this could affect the analysis. These two positions are contradictory. Either there is significant tipping, in which case it must be included in the analysis, or there isn't, in which case lines 384-386 should say something different.

Reviewer 3 Report

Thanks to the authors for making these changes and responding to each of my comments. I am satisfied with the responses to my comments and think has strengthened the manuscript substantially. I just have two further minor comments below.

However, I am concerned that it appears that the researchers did not go through a formal process to seek and obtain ethical clearance to conduct this research. Although informed consent from participants is essential, my feeling is that clearance/approval from an ethical review board is necessary given that the research involved vulnerable populations in discussing a fairly sensitive topic. I think it is up to the editors if they wish to publish research that has not gone through rigorous ethical review, but I personally am not comfortable with this. IJERPH has its own statement on this as well: https://www.mdpi.com/journal/ijerph/instructions#rethics

Lines 110-113: I still think more detail on the interviews should be provided. How many individuals from each of the organisations were interviewed and what interview topics were covered?

Lines 358-359: Thanks to authors for clarifying in their responses what the source of these assumptions are. I think it would be helpful for the authors to include this explanation in the text (that these assumptions were derived from the interviews with government officials).
